# Variable Structure PID Controller for Satellite Attitude Control Considering Actuator Failure

**Yong Qi** [1,2,*] , **Haizhao Jing** [2] and **Xiwei Wu** [3]

1    School of Electronic Information and Artificial Intelligence, Shaanxi University of Science and Technology, Xi'an 710021, China
2    Shaanxi Joint Laboratory of Artificial Intelligence, Shaanxi University of Science and Technology, Xi'an 710021, China; 201915030414@sust.edu.cn
3    School of Automation, Northwestern Polytechnical University, Xi'an 710129, China; wxw-treffen@mail.nwpu.edu.cn
*    Correspondence: qiyong_sust@163.com

**Abstract:** In this paper, a variable structure PID controller with a good convergence rate and robustness for satellite attitude is proposed. In order to improve the system convergence rate, the variable structure for the proportional and differential term was designed, and an angular velocity curve with a better convergence rate was achieved by this variable structure. In addition, an integral partitioning algorithm was designed, and the system robustness to disturbance torque was improved; meanwhile, the negative effect of the integral term was avoided during the converging process. The actuator failure condition was also considered, and a fault tolerant control algorithm was designed. System stability was analyzed by the Lyapunov method, and its performance was demonstrated by numerical simulation.

**Keywords:** satellite attitude control; PID control; fault tolerant; actuator failure





## 1. Introduction

With the rapid development of the aerospace industry, more and more attention has been paid to the high precision attitude control for satellites. To ensure high precision, high reliability, and a fast convergence rate of the satellite, the control system of the satellite must have sufficient control capability. At present, many researchers have studied the basic theories and problems of satellite attitude control [1–6]. As a commonly used method in satellite attitude control, PID control can achieve the closed-loop control system, reaching a steady state, and it has the advantages of a simple structure and easy engineering implementation. However, the control accuracy of PID control is relatively low, which often cannot meet the requirements of current missions. Furthermore, in order to ensure the stability of the satellite attitude control system under actuator failure (which is quite often in space missions and is one of the major reasons for mission failures), it is necessary to enhance the system fault tolerant capability to actuator failures. Hence, it is necessary to modify current PID controllers to meet the precision, reliability, and fast convergence rate requirements.

For the PID control method, Zhang [7] designed a low-complexity structured H-infinity controller for flexible satellite attitude control considering interference suppression. Zhang [8] designed an adaptive sliding mode integral controller for the control of the robotic arm of the spacecraft and used a PID-type integral sliding mode control surface to reduce the steady-state error. Li [9] designed a fuzzy PID control algorithm according to the characteristics of the three-axis stable zero-momentum wheel. The authors used a PID control algorithm to manipulate the satellite attitude, but the result could not achieve the desired control accuracy. In summary, researchers made some improvements to the PID

control algorithm, but there is still a lack of study on a fault-tolerant control algorithm with a good convergence rate, combined with a PID control algorithm.

For the fault-tolerant control method of satellite attitude control, Zhang [10] designed a satellite attitude fault-tolerant control algorithm for an agile satellite attitude control system, which can achieve agile satellite attitude control and accurately output control torque, with the purpose of fast maneuver and high-precision stability attitude under fault conditions. Li [11] designed an adaptive fault-tolerant attitude tracking control algorithm based on a neural network for the case of partial failure or even complete failure of the spacecraft actuator. Chamitoff [12] proposed an adaptive fault-tolerant control method based on a disturbance observer for the problems of actuator failure and space environment disturbance in a flexible satellite attitude control system. These works only focus on the problem of satellite control system fault tolerance and anti-jamming ability, but they do not pay attention to the system convergence rate.

A variable structure controller is one of the main methods to improve the system convergence rate. Cai [13] designed a two-stage pulse variable structure phase plane partition attitude controller for an integrated control system of a navigation satellite four-inclined chemical propulsion. Chen [14] designed a variable structure controller based on error quaternion and error angular velocity for the large-angle attitude maneuver problem of agile satellites. For the control problem of satellite attitude, a variable structure control method is used to achieve higher precision control compared with PID control, but most works do not pay attention to the fault tolerant issue.

In addition, the PID fault-tolerant control method has been applied to the field of electric vehicles [15], servo motors [16], quadcopter drones [17], and so on. For the variable structure PID control method, it can also be applied to stepper motors [18], robots [19], and so on. System performance is largely improved by implementing the proposed method. In addition, Li [20–22] and Xiao [23–26] designed satellite attitude controllers for fast attitude maneuver, but the actuator failures were not taken into consideration. Hence, in this paper, a variable structure PID controller with fault tolerant capability was proposed for satellite attitude control to improve the system convergence rate and robustness.

## 2. Methodology

The dynamic model of the flexible satellite using the Euler theorem under the hypothesis of small elastic deformations could be written as follows.

$$J\dot{\omega} = -\omega^\times J\omega + u + d \tag{1}$$

where $J$ is the inertia matrix of the satellite, which is a symmetric positive definite matrix. Generally, $J$ cannot be accurately known and could be written as $J = \hat{J} + \tilde{J}$, where $\hat{J}$ is the inertia matrix best estimate and $\tilde{J}$ is the error inertia matrix.

In Equation (1), $d$ is the unknown disturbance torque with norm upper bound $\|d\| \leq \bar{d}$, and $u$ is the control torque to be designed. The product matrix $r^\times$ of the three-dimensional vector $r$ is defined as follows:

$$r^\times = \begin{bmatrix} 0 & -r_3 & r_2 \\ r_3 & 0 & -r_1 \\ -r_2 & r_1 & 0 \end{bmatrix} \tag{2}$$

The product matrix $r^\times$ has an important property, which will be used in a later part of this paper, that the eigenvalues of $r^\times$ satisfies

$$\begin{aligned} \lambda(r^\times) &= 0, \|r\|_2 \\ \lambda_{\max}(r^\times) &= \|r\|_2 \end{aligned} \tag{3}$$

In order to simplify the text, the maximum and minimum eigenvalue of matrix $A$ is described as $\lambda_M(A), \lambda_m(A)$.

The kinetic model of the satellite could be written as follows:

$$\dot{\boldsymbol{q}} = \begin{bmatrix} \dot{q}_0 \\ \dot{\boldsymbol{q}}_v \end{bmatrix} = \begin{bmatrix} -\frac{1}{2}\boldsymbol{q}_v^T\boldsymbol{\omega} \\ \frac{1}{2}(q_0\boldsymbol{I}_3 + \boldsymbol{q}_v^\times)\boldsymbol{\omega} \end{bmatrix} = \frac{1}{2}\begin{bmatrix} -\boldsymbol{q}_v^T \\ \boldsymbol{F} \end{bmatrix}\boldsymbol{\omega} \tag{4}$$

where $\boldsymbol{F} = q_0\boldsymbol{I}_3 + \boldsymbol{q}_v^\times$, and the eigenvalue of $\boldsymbol{F}$ satisfies

$$\begin{aligned} \lambda(\boldsymbol{F}) &= |q_0|, 1 \\ \lambda_M(\boldsymbol{F}) &= 1 \end{aligned} \tag{5}$$

In addition, the Euler angle/axis description for satellite attitude is widely used, and its kinetic model is given as follows:

$$\begin{cases} \dot{\boldsymbol{e}} = \frac{1}{2}\boldsymbol{e}^\times\left(\boldsymbol{I}_3 - \cot\frac{\varphi}{2}\boldsymbol{e}^\times\right)\boldsymbol{\omega} \\ \dot{\varphi} = \boldsymbol{e}^T\boldsymbol{\omega} \end{cases} \tag{6}$$

where $\boldsymbol{e}$ is defined as the Euler axis, and $\varphi$ is defined as the Euler angle.

The next two assumptions used in this paper are introduced as follows.

**Assumption 1.** *It is assumed that the disturbance torque $\boldsymbol{d}$ is the norm upper bounded, i.e.,*

$$\|\boldsymbol{d}\| \leq \bar{d} \tag{7}$$

*where $\bar{d}$ is a positive scalar.*

**Assumption 2.** *It is assumed that, in this paper, if matrix $\boldsymbol{A}$ satisfies $\boldsymbol{A} \geq 0$, it means that matrix $\boldsymbol{A}$ is a positive definite matrix, i.e., $\boldsymbol{x}^T\boldsymbol{A}\boldsymbol{x} \geq 0$ holds for any non-zero vector $\boldsymbol{x}$.*

## 3. Results

### 3.1. Variable Structure PD Controller

In the field of satellite attitude control, the PD control algorithm is a mature and widely used method since its simple structure and satellite attitude system governed by the PD controller could be treated as a second order system. The main drawback of a PD controller could be concluded as the following two aspects: system convergence rate is relatively low compared with current control methods, such as finite time control and fixed time control; the system anti-perturbation capability is also relatively low, and the system is sensitive to disturbance torques. In order to deal with the first issue, a common method involves enlarging the control gain parameters, which could aggravate the second issue. In addition, a common method to deal with the second issue is to add an integral term to the PD controller, i.e., design a PID control algorithm, which could slow down the system convergence rate and aggravate the first issue. Based on the discussion above, it could be found that these issues are deeply coupled and are hard to solve based on standard methods. In this section the variable structure control method is implemented, and a variable structure PD controller will be designed to improve the system convergence rate when the system state is away from its equilibrium point.

The proposed variable structure PD controller could be written as follows:

$$\boldsymbol{u} = \begin{cases} -k_d\boldsymbol{\omega} - k_p\boldsymbol{q}_v - \bar{d}sign(\boldsymbol{\omega}) + \boldsymbol{\tau} & \|\boldsymbol{q}_v\| \geq \alpha \\ -k_d\boldsymbol{\omega} - k_p\boldsymbol{q}_v - \bar{d}sign(\boldsymbol{\omega}) & \|\boldsymbol{q}_v\| < \alpha \end{cases} \tag{8}$$

where $\alpha$ is a positive scalar whose function is to detect the system state as it approaches its equilibrium point, $\bar{d}$ is the norm upper bound of disturbance torque, and $k_d$ and $k_p$ are also positive control parameters, defined as follows:

$$k_d = \text{constant}$$
$$k_p = \begin{cases} k_1 k_d / \|q_v\| & \|q_v\| \geq \alpha \\ \text{constant} & \|q_v\| < \alpha \end{cases} \tag{9}$$

where $k_1$ is a positive scalar. The additional control term $\tau$ in (8) is defined as follows:

$$\tau = \omega^\times \left( \hat{J}\omega - \tfrac{1}{2}\hat{J}\eta - \tfrac{1}{2} \cot \tfrac{\varphi}{2} e^\times \hat{J}\eta \right) \tag{10}$$
$$\eta = \omega + k_1 e$$

**Remark 1.** *Based on the structure of the controller (8), it could be found that the system has two stages: (1) when the system state is far away from its equilibrium point, i.e., when $\|q_v\| \geq \alpha$, the proportion term enlarges as the system state converges, and an additional term with angular velocity produce matrix is added (its function will be introduced in later text), and during this stage the controller is structure-variable; (2) when the system state nears its equilibrium point, the controller (8) transforms to a standard PD controller, and the system has a simple structure.*

**Remark 2.** *The proposed controller (8) has the following properties, which will be proved in later text: 1. system (1) and (4) governed by controller (8) are globally asymptotic stable, i.e., with all positive scalars $k_d$, $k_p$, and $k_1$, the system is asymptotic stable; 2. during the first stage of controller (8), the system angular velocity is nearly constant, i.e., $\eta \approx 0$, and the system could have a better convergence rate compared with a standard PD controller.*

The next step is to prove the properties described in Remark 2. First, system stability is discussed. The Lyapunov function is selected as follows:

$$V_1 = \frac{1}{2}\omega^T J\omega + 2k(1 - q_0) \tag{11}$$

where $k$ is a positive scalar.

During the first stage of controller (8), the derivative of (11) can be obtained by

$$\begin{aligned}
\dot{V}_1 &= \omega^T J\dot{\omega} - 2k_p \dot{q}_0 \\
&= \omega^T \left( -k_d \omega - k_p q_v - \bar{d}\,sign(\omega) + \tau + d - \omega^\times J\omega \right) \\
&= -k_d \omega^T \omega + \omega^T \omega^\times \left( \hat{J}\omega - \tfrac{1}{2}\hat{J}\eta - \tfrac{1}{2} \cot \tfrac{\varphi}{2} e^\times \hat{J}\eta \right) + \omega^T d - \bar{d}\omega^T sgn(\omega) \\
&= -k_d \omega^T \omega + \omega^T d - \bar{d}\omega^T sgn(\omega) \\
&\leq 0
\end{aligned} \tag{12}$$

In addition, during the second stage of controller (8), the derivative of (11) can be obtained by

$$\begin{aligned}
\dot{V}_1 &= \omega^T J\dot{\omega} - 2k_p \dot{q}_0 \\
&= -k_d \omega^T \omega + \omega^T d - \bar{d}\omega^T sgn(\omega) \\
&\leq 0
\end{aligned} \tag{13}$$

Hence, the system global asymptotic stability is proved.

The next step is to discuss the system convergence rate under controller (8). The Lyapunov function is selected as follows:

$$V_2 = \frac{1}{2}\eta^T J\eta \tag{14}$$

During the first stage of controller (8), the derivative of $\eta$ can be obtained by

$$
\begin{aligned}
\dot{V}_2 &= \eta^T J \dot{\eta} \\
&= \eta^T \left( J\dot{\omega} + Jk_1\dot{e} \right) \\
&= \eta^T \left( u + d - \omega^\times J\omega + \tfrac{1}{2}k_1 Je^\times \left( I_3 - \cot \tfrac{\varphi}{2} e^\times \right) \omega \right) \\
&\approx -k_d \eta^T \eta \leq 0
\end{aligned}
\tag{15}
$$

Thus, auxiliary state $\eta$ is also asymptotic stable, i.e., $\eta \to 0$, and that means that during this stage $\omega = -k_1 e$. Hence, the system could have a constant convergence rate during this stage, and by implementing this variable structure controller, the system convergence rate could be improved. The properties in Remark 2 have been proven.

*3.2. Integral Partition Design*

Although the PD controller for satellite attitude control has such advantages as discussed above, one of its main drawbacks is its low robustness to disturbances, especially high frequency random noise. Noting that this type of noise is quite common in satellite engineering practice (always caused by electric motors), and this type of noise is hard to measure and resist, it is necessary to take this perturbation into consideration when designing satellite attitude controllers. In engineering practice, a PID controller with an integral term is an effective method to resist this type of perturbation, but the integral term slow down system convergence rate and system is unstable when the integral term is not designed properly.

Based on the discussion above, the idea to design a variable structure PID controller could be concluded as follows: (1) when the system state is away from its equilibrium point, the controller maintains a PD controller, as stated in the previous section, to improve the system convergence rate, and the drawback of an integral term such as a low convergence rate could be avoided; (2) noting that the influence of perturbations mainly exists when the system state approaches its equilibrium point, the integral term is added into the controller to resist the influence of disturbance torque; (3) the main difficulty of designing a PID controller is analyzing the system stability, and noting that the condition of the integral term is that the system nearly reaches its equilibrium point, some assumptions could be made when analyzing system stability.

The variable structure PID controller proposed in this paper could be written as follows:

$$
u = \begin{cases}
-k_d\omega - k_p q_v - \overline{d}\,sign(\omega) + \tau & \|q_v\| \geq \alpha \\
-k_d\omega - k_p q_v - \overline{d}\,sign(\omega) & \beta \leq \|q_v\| < \alpha \\
-k_d\omega - k_p q_v - k_I v & \|q_v\| < \beta
\end{cases}
\tag{16}
$$

where $k_d, k_p, k_I$, and $\alpha, \beta$ are all positive scalars, the definition of control term $\tau$ is totally same as Equation (10), and the definition of integral term $v$ is given as follows:

$$
\dot{v} = q_v, v(0) = 0_{3\times 1}
\tag{17}
$$

**Remark 3.** *Based on the controller structure of (16), it could be found that this controller consists of three stages; the first two stages are totally the same as variable structure PD controller (8), to improve system performance when the system state is relatively large, and the third stage is the integral partition term to resist disturbance when the system state approaches its equilibrium point. It could also be found that there is no sign function term during this stage to avoid the high frequency vibration of system state. In addition, the parameters $\alpha$ and $\beta$ are designed to detect whether the system state is near zero or not, and the selection principle could be made depending on the system capability and control mission. The larger torque the actuator can generate and the faster a convergence rate is needed, a smaller $\alpha$ and $\beta$ should be selected, and vice versa.*

The next step is to discuss system stability under controller (16). Noting that the first two stages are totally the same as controller (8), and system stability during these stages has been proven in the previous section, this section only discusses the stability during the third stage. The Lyapunov function is selected as follows:

$$V_2 = \frac{1}{2}\boldsymbol{\omega}^T \boldsymbol{J}\boldsymbol{\omega} + l_1 \boldsymbol{q}_v^T \boldsymbol{J}\boldsymbol{\omega} + 2l_2(1 - q_0) + \frac{1}{2}l_3 \boldsymbol{v}^T \boldsymbol{v} + l_4 \boldsymbol{v}^T \boldsymbol{J}\boldsymbol{\omega} \tag{18}$$

where $l_1, l_2, l_3, l_4$ are all positive scalars.

Noting that $-1 \leq q_0 \leq 1$, we have

$$1 - q_0 \geq \frac{1}{2}(1 + q_0)(1 - q_0) = \frac{1}{2}\left(1 - q_0^2\right) = \boldsymbol{q}_v^T \boldsymbol{q}_v \tag{19}$$

and

$$V_2 \geq \frac{1}{2}\boldsymbol{\omega}^T \boldsymbol{J}\boldsymbol{\omega} + l_1 \boldsymbol{q}_v^T \boldsymbol{J}\boldsymbol{\omega} + l_2 \boldsymbol{q}_v^T \boldsymbol{q}_v + \frac{1}{2}l_3 \boldsymbol{v}^T \boldsymbol{v} + l_4 \boldsymbol{v}^T \boldsymbol{J}\boldsymbol{\omega}$$

$$= \begin{bmatrix} \boldsymbol{\omega} \\ \boldsymbol{q}_v \\ \boldsymbol{v} \end{bmatrix}^T \begin{bmatrix} \frac{1}{2}\boldsymbol{J} & \frac{1}{2}l_1\boldsymbol{J} & \frac{1}{2}l_4\boldsymbol{J} \\ \frac{1}{2}l_1\boldsymbol{J} & l_2\boldsymbol{I}_3 & 0 \\ \frac{1}{2}l_4\boldsymbol{J} & 0 & \frac{1}{2}l_3\boldsymbol{I}_3 \end{bmatrix} \begin{bmatrix} \boldsymbol{\omega} \\ \boldsymbol{q}_v \\ \boldsymbol{v} \end{bmatrix} \tag{20}$$

If following inequality is satisfied, then the Lyapunov function is a positive definite.

$$2l_2\boldsymbol{I}_3 - l_1^2\boldsymbol{J} \geq 0$$
$$2l_2l_3\boldsymbol{I}_3 - 2l_2l_4^2\boldsymbol{J} - l_1^2l_3\boldsymbol{J} \geq 0 \tag{21}$$

The derivative of the Lyapunov function in (18) is calculated and the third stage of controller (16) is substituted, so that the following can be obtained:

$$\dot{V}_2 = \boldsymbol{\omega}^T \boldsymbol{J}\dot{\boldsymbol{\omega}} + l_1 \boldsymbol{q}_v^T \boldsymbol{J}\dot{\boldsymbol{\omega}} + l_1 \boldsymbol{\omega}^T \boldsymbol{J}\dot{\boldsymbol{q}}_v - 2l_2\dot{q}_0 + l_3 \boldsymbol{v}^T \dot{\boldsymbol{v}} + l_4 \boldsymbol{v}^T \boldsymbol{J}\dot{\boldsymbol{\omega}} + l_4 \boldsymbol{\omega}^T \boldsymbol{J}\dot{\boldsymbol{v}}$$
$$= (\boldsymbol{\omega} + l_1 \boldsymbol{q}_v + l_3 \boldsymbol{v})^T (\boldsymbol{u} + \boldsymbol{d} - \boldsymbol{\omega}^\times \boldsymbol{J}\boldsymbol{\omega}) + l_1 \boldsymbol{\omega}^T \boldsymbol{J}\boldsymbol{F}\boldsymbol{\omega} + l_2 \boldsymbol{\omega}^T \boldsymbol{q}_v + l_3 \boldsymbol{v}^T \boldsymbol{q}_v + l_4 \boldsymbol{\omega}^T \boldsymbol{q}_v \tag{22}$$

Noting that, during this stage, the system state is near its equilibrium point, the high-order term $\boldsymbol{\omega}^\times \boldsymbol{J}\boldsymbol{\omega}$ is ignored. In order to analyze system stability, disturbance torque $\boldsymbol{d}$ is considered in later text, and we have:

$$\dot{V}_2 \approx (\boldsymbol{\omega} + l_1 \boldsymbol{q}_v + l_3 \boldsymbol{v})^T \boldsymbol{u} + l_1 \boldsymbol{\omega}^T \boldsymbol{J}\boldsymbol{F}\boldsymbol{\omega} + l_2 \boldsymbol{\omega}^T \boldsymbol{q}_v + l_3 \boldsymbol{v}^T \boldsymbol{q}_v + l_4 \boldsymbol{\omega}^T \boldsymbol{q}_v$$
$$= -(\boldsymbol{\omega} + l_1 \boldsymbol{q}_v + l_3 \boldsymbol{v})(k_d \boldsymbol{\omega} + k_p \boldsymbol{q}_v + k_I \boldsymbol{v}) + l_1 \boldsymbol{\omega}^T \boldsymbol{J}\boldsymbol{F}\boldsymbol{\omega} + l_2 \boldsymbol{\omega}^T \boldsymbol{q}_v + l_3 \boldsymbol{v}^T \boldsymbol{q}_v + l_4 \boldsymbol{\omega}^T \boldsymbol{q}_v$$
$$\leq - \begin{bmatrix} \boldsymbol{\omega} \\ \boldsymbol{q}_v \\ \boldsymbol{v} \end{bmatrix}^T \begin{bmatrix} k_d \boldsymbol{I}_3 - l_1 \boldsymbol{J} & \frac{1}{2}\boldsymbol{P} & \frac{1}{2}(k_I + l_4 k_d)\boldsymbol{I}_3 \\ \frac{1}{2}\boldsymbol{P} & l_1 k_p \boldsymbol{I}_3 & \frac{1}{2}(l_1 k_I + l_4 k_p - l_3)\boldsymbol{I}_3 \\ \frac{1}{2}(k_I + l_4 k_d)\boldsymbol{I}_3 & \frac{1}{2}(l_1 k_I + l_4 k_p - l_3)\boldsymbol{I}_3 & l_4 k_I \boldsymbol{I}_3 \end{bmatrix} \begin{bmatrix} \boldsymbol{\omega} \\ \boldsymbol{q}_v \\ \boldsymbol{v} \end{bmatrix} \tag{23}$$

where $\boldsymbol{P} = (l_1 k_d + k_p l_4 - l_2)\boldsymbol{I}_3 - l_4 \boldsymbol{J}$.

Hence, if following inequality is satisfied, $\dot{V}_2 \leq 0$ could be ensured and the system governed by controller (16) is asymptotic stable.

$$\begin{bmatrix} k_d \boldsymbol{I}_3 - l_1 \boldsymbol{J} & \frac{1}{2}\boldsymbol{P} & \frac{1}{2}(k_I + l_4 k_d)\boldsymbol{I}_3 \\ \frac{1}{2}\boldsymbol{P} & l_1 k_p \boldsymbol{I}_3 & \frac{1}{2}(l_1 k_I + l_4 k_p - l_3)\boldsymbol{I}_3 \\ \frac{1}{2}(k_I + l_4 k_d)\boldsymbol{I}_3 & \frac{1}{2}(l_1 k_I + l_4 k_p - l_3)\boldsymbol{I}_3 & l_4 k_I \boldsymbol{I}_3 \end{bmatrix} \geq 0 \tag{24}$$

Noting that this inequality consists of several control parameters and matrices, and none of the elements of the matrix in (24) is zero, it is hard to examine (24) manually. Noting that some of the coupled terms could be simplified, the control parameters are selected as follows:

$$l_2 = l_1 k_d + k_p$$
$$l_3 = l_1 k_I + l_4 k_p \tag{25}$$

and the derivative of the Lyapunov function can be simplified as:

$$\dot{V}_2 = -\boldsymbol{\omega}^T(k_d\boldsymbol{I}_3 - l_1\boldsymbol{JF})\boldsymbol{\omega} - l_1 k_p \boldsymbol{q}_v^T \boldsymbol{q}_v - l_4 k_I \boldsymbol{v}^T \boldsymbol{v} - l_4 \boldsymbol{q}_v^T \boldsymbol{J}\boldsymbol{\omega} - (k_I + l_4 k_d)\boldsymbol{\omega}^T \boldsymbol{v}$$

$$\leq - \begin{bmatrix} \boldsymbol{\omega} \\ \boldsymbol{q}_v \\ \boldsymbol{v} \end{bmatrix}^T \begin{bmatrix} k_d\boldsymbol{I}_3 - l_1\boldsymbol{J} & \frac{1}{2}l_4\boldsymbol{J} & \frac{1}{2}(k_I + l_4 k_d)\boldsymbol{I}_3 \\ \frac{1}{2}l_4\boldsymbol{J} & l_1 k_p\boldsymbol{I}_3 & 0 \\ \frac{1}{2}(k_I + l_4 k_d)\boldsymbol{I}_3 & 0 & l_4 k_I\boldsymbol{I}_3 \end{bmatrix} \begin{bmatrix} \boldsymbol{\omega} \\ \boldsymbol{q}_v \\ \boldsymbol{v} \end{bmatrix} \tag{26}$$

Hence, if the following inequality is satisfied, then $\dot{V}_2 \leq 0$, and the system is asymptotic stable:

$$4l_1 k_p(k_d\boldsymbol{I}_3 - l_1\boldsymbol{J}) - l_4^2\boldsymbol{J}^2 > 0$$
$$4l_1 l_4 k_p k_I(k_d\boldsymbol{I}_3 - l_1\boldsymbol{J}) - l_4^2\boldsymbol{J}^2 - (l_2 + l_4 k_d)\boldsymbol{I}_3 > 0 \tag{27}$$

Based on the discussion above, the constraints on control parameters could be concluded as follows:

$$\begin{cases} 2(l_1 k_d + k_p)\boldsymbol{I}_3 - l_1^2\boldsymbol{J} \geq 0 \\ 2(l_1 k_d + k_p)(l_1 k_I + l_4 k_p)\boldsymbol{I}_3 - 2(l_1 k_d + k_p)l_4^2\boldsymbol{J} - l_1^2(l_1 k_I + l_4 k_p)\boldsymbol{J} \geq 0 \\ 4l_1 k_p(k_d\boldsymbol{I}_3 - l_1\boldsymbol{J}) - l_4^2\boldsymbol{J}^2 > 0 \\ 4l_1 l_4 k_p k_I(k_d\boldsymbol{I}_3 - l_1\boldsymbol{J}) - l_4^2\boldsymbol{J}^2 - (l_1 k_d + k_p + l_4 k_d)\boldsymbol{I}_3 > 0 \end{cases} \tag{28}$$

It could be found that, in constraint (28), there are some matrix calculations that are not convenient for manual calculation, and (28) could be further transformed to:

$$\begin{cases} 2(l_1 k_d + k_p) - l_1^2\lambda_M(\boldsymbol{J}) \geq 0 \\ 2(l_1 k_d + k_p)(l_1 k_I + l_4 k_p) - 2(l_1 k_d + k_p)l_4^2\lambda_M(\boldsymbol{J}) - l_1^2(l_1 k_I + l_4 k_p)\lambda_M(\boldsymbol{J}) \geq 0 \\ 4l_1 k_p(k_d - l_1\lambda_M(\boldsymbol{J})) - l_4^2\lambda_M^2(\boldsymbol{J}) > 0 \\ 4l_1 l_4 k_p k_I(k_d - l_1\lambda_M(\boldsymbol{J})) - l_4^2\lambda_M^2(\boldsymbol{J}) - (l_1 k_d + k_p + l_4 k_d) > 0 \end{cases} \tag{29}$$

**Remark 4.** *Both (28) and (29) are strict constraints on control parameters, which means that if any inequality is satisfied, the system asymptotic stability can be ensured. The difference between (28) and (29) is that (28) involves some matrix calculation and (29) is simply a scalar calculation. In addition, (29) is stricter than (28), which means that if (29) is satisfied, then (28) holds.*

In addition, noting that there exists unknown disturbance torque, the system state cannot stay at the equilibrium point but must pass through the equilibrium point continuously; hence, it is necessary to discuss the steady accuracy related to disturbance. Considering the disturbance torque, the derivative of the Lyapunov function can be re-calculated as:

$$\begin{aligned} \dot{V}_2 &= -\boldsymbol{\omega}^T(k_d\boldsymbol{I}_3 - l_1\boldsymbol{JF})\boldsymbol{\omega} - l_1 k_p \boldsymbol{q}_v^T \boldsymbol{q}_v - l_4 k_I \boldsymbol{v}^T \boldsymbol{v} - l_4 \boldsymbol{q}_v^T \boldsymbol{J}\boldsymbol{\omega} - (k_I + l_4 k_d)\boldsymbol{\omega}^T \boldsymbol{v} \\ &\quad + (\boldsymbol{\omega} + l_1 \boldsymbol{q}_v + l_3 \boldsymbol{v})^T \boldsymbol{d} \\ &\leq -(k_d - l_1\lambda_M(\boldsymbol{J}))\|\boldsymbol{\omega}\|^2 - l_1 k_p\|\boldsymbol{q}_v\|^2 - l_4 k_I\|\boldsymbol{v}\|^2 + l_4\lambda_M(\boldsymbol{J})\|\boldsymbol{q}_v\|\|\boldsymbol{\omega}\| \\ &\quad + (k_I + l_4 k_d)\|\boldsymbol{\omega}\|\|\boldsymbol{v}\| + (\|\boldsymbol{\omega}\| + l_1\|\boldsymbol{q}_v\| + l_3\|\boldsymbol{v}\|)\bar{d} \\ &\leq -(k_d - l_1\lambda_M(\boldsymbol{J}))\|\boldsymbol{\omega}\|^2 - l_1 k_p\|\boldsymbol{q}_v\|^2 - l_4 k_I\|\boldsymbol{v}\|^2 + \frac{1}{2}l_4\lambda_M(\boldsymbol{J})\left(\|\boldsymbol{\omega}\|^2 + \|\boldsymbol{q}_v\|^2\right) \\ &\quad + \frac{1}{2}(k_I + l_4 k_d)\left(\|\boldsymbol{\omega}\|^2 + \|\boldsymbol{v}\|^2\right) + (\|\boldsymbol{\omega}\| + l_1\|\boldsymbol{q}_v\| + l_3\|\boldsymbol{v}\|)\bar{d} \\ &= -\left((k_d - l_1\lambda_M(\boldsymbol{J})) - \frac{1}{2}l_4\lambda_M(\boldsymbol{J}) - \frac{1}{2}(k_I + l_4 k_d)\right)\|\boldsymbol{\omega}\|^2 + \bar{d}\|\boldsymbol{\omega}\| \\ &\quad - \left(l_1 k_p - \frac{1}{2}l_4\lambda_M(\boldsymbol{J})\right)\|\boldsymbol{q}_v\|^2 + \bar{d}\|\boldsymbol{q}_v\| - \left(l_4 k_I - \frac{1}{2}(k_I + l_4 k_d)\right)\|\boldsymbol{v}\|^2 + \bar{d}\|\boldsymbol{v}\| \end{aligned} \tag{30}$$

Hence, it can be found that, if the following inequality is satisfied, then $\dot{V}_2 \leq 0$ can be ensured.

$$\|\boldsymbol{\omega}\| \geq \frac{\bar{d}}{(k_d - l_1\lambda_M(\boldsymbol{J})) - \frac{1}{2}l_4\lambda_M(\boldsymbol{J}) - \frac{1}{2}(k_I + l_4 k_d)}$$
$$\|\boldsymbol{q}_v\| \geq \frac{\bar{d}}{l_1 k_p - \frac{1}{2}l_4\lambda_M(\boldsymbol{J})}, \|\boldsymbol{v}\| \geq \frac{\bar{d}}{l_4 k_I - \frac{1}{2}(k_I + l_4 k_d)} \tag{31}$$

In other words, if inequality (31) is satisfied, the system can converge to its equilibrium point, and it can be concluded that, under disturbance torque, the system steady accuracy satisfies:

$$\|\boldsymbol{\omega}\| < \frac{\overline{d}}{(k_d - l_1 \lambda_M(\boldsymbol{J})) - \frac{1}{2}l_4 \lambda_M(\boldsymbol{J}) - \frac{1}{2}(k_I + l_4 k_d)}, \|\boldsymbol{q}_v\| < \frac{\overline{d}}{l_1 k_p - \frac{1}{2}l_4 \lambda_M(\boldsymbol{J})} \quad (32)$$

**Remark 5.** *The steady accuracy field (32) is not strict since some in-equal transformations are made during the derivation. The system state may converge to a smaller field than (32) and stay within it, and inequality (32) is the steady accuracy considering the worst condition. It can be found that there are many control parameters in the stability analysis, but there are only three parameters in the controller, i.e., $k_d, k_p, k_I$, and other parameters are selected to analyze system stability, but they do not affect system performance.*

### 3.3. Fault Tolerant Algorithm

Noting that the actuators on the satellite platform may fail during the control process and cause the divergence of the control system, it is necessary to design a fault detect and a tolerant algorithm for the satellite attitude controller. It is worth noting that, under a complete failure situation (actuators could not generate any torque at all), the system is completely out of control, and the control algorithm would not help for hardware breakdown. In this section, the part failure is considered, i.e., control actuators would generate part of the expected torque as follows:

$$\boldsymbol{u} = \boldsymbol{\rho} \boldsymbol{u}_d \quad (33)$$

where $\boldsymbol{u}$ is the real torque generated by the actuator, $\boldsymbol{u}_d$ is the expected torque calculated by the control algorithm, and $\boldsymbol{\rho}$ is the error matrix describing the coupling of real torque and expected torque; generally, the element of $\boldsymbol{\rho}$ satisfies $|\rho_{ij}| \leq 1$.

Based on the results in previous section, the fault tolerant variable structure PID controller could be written as follows

$$\boldsymbol{u}_r = \begin{cases} \boldsymbol{u} & \delta \leq \overline{\delta} \\ -k_d \boldsymbol{\omega} - k_p \boldsymbol{q}_v & \delta > \overline{\delta} \end{cases} \quad (34)$$

where $\boldsymbol{u}_r$ is the fault tolerant control torque, $\boldsymbol{u}$ is the control torque (same as the previous section, i.e., Equation (16)), $k_d$, $k_p$, and $\overline{\delta}$ are positive scalars, and $\delta$ is the fault detection factor that detects whether the fault happens. The definition of $\delta$ is defined as follows:

$$\delta = \|\boldsymbol{\omega} - \boldsymbol{\omega}_{k-1} - \boldsymbol{J}^{-1}\left(\boldsymbol{u}_r - \boldsymbol{\omega}_{k-1}^{\times} \boldsymbol{J} \boldsymbol{\omega}_{k-1}\right)\Delta t\| \quad (35)$$

where $\boldsymbol{\omega}_{k-1}$ is the angular velocity of the last control cycle, and $\Delta t$ is the control sample time.

It could be easily found that the later part in Equation (35) is integral to the dynamic model of angular velocity of the last control cycle, and the former part is the current angular velocity; hence, the essence of $\delta$ is to calculate the error of the estimated and actual values.

**Remark 6.** *Based on the structure of the fault tolerant controller, it can be found that the key to detect the fault is $\delta$, and the basic idea is to integrate the dynamic model of angular velocity of the last control cycle and to compare it with the current angular velocity. If the theoretical value is too much larger than the threshold $\overline{\delta}$, then it could be treated as a system with actuator failure. In addition, based on the controller structure, it could be found that if the actuator failure does not occur, then the controller maintains the same structure as the previous section, and if the failure does happen, the controller becomes a standard PD controller.*

The next step is to prove system stability under controller (34). Since the stability without actuator failure has been proven in the previous section, only the fault situation is discussed. The Lyapunov function is selected as follows

$$V_3 = \frac{1}{2}\boldsymbol{\omega}^T\boldsymbol{J}\boldsymbol{\omega} + 2k_p(1 - q_0) \tag{36}$$

Its derivative is calculated and the second stage of controller (34) is substituted

$$\begin{aligned}\dot{V}_3 &= \boldsymbol{\omega}^T\boldsymbol{J}\dot{\boldsymbol{\omega}} - 2k_p\dot{q}_0 \\ &\approx \boldsymbol{\omega}^T(\boldsymbol{u} - \boldsymbol{\omega}^\times\boldsymbol{J}\boldsymbol{\omega}) + k_p\boldsymbol{q}_v^T\boldsymbol{\omega} \\ &= -k_d\boldsymbol{\omega}^T\boldsymbol{\omega} \le 0\end{aligned} \tag{37}$$

Based on the LaSalle invariant principle, the system converges to the curve $\boldsymbol{\omega} = 0$, hence $\dot{\boldsymbol{\omega}} = 0$; noting the dynamic model of $\dot{\boldsymbol{\omega}}$ and the controller structure, it could be obtained that $\boldsymbol{q}_v = 0$. Hence, the system governed by controller (34) is asymptotic stable.

**Remark 7.** *Based on the stability analysis above, it could be found that the PD controller has an important property that, for any positive parameter $k_p$ and $k_d$, it is the key to solving the actuator failure issue. Noting that the fault situation is the output control torque being cut down, and it could be treated as that, the controller is generated by another positive control parameter. Hence, this method only effects the system performance parameters such as convergence rate and steady accuracy, but system stability could be ensured.*

**Remark 8.** *The essence to solve actuator failure is a property of the PD controller. Global stability means that, even if part of the control torque is lost in the actuator, this condition could be treated, as a new group of control gain parameters can be implemented, and system stability could be ensured. For example, if the origin control parameters are $k_p$ and $k_d$, and when actuator failure occurs and its coupling matrix is $\boldsymbol{\rho}$, it could be treated as if the new Lyapunov function is*

$$V_\rho = \frac{1}{2}\boldsymbol{\omega}^T\boldsymbol{J}\boldsymbol{\omega} + 2\boldsymbol{\rho}k_p(1 - q_0) \tag{38}$$

Its derivative is calculated, and the controller $\boldsymbol{u}_r = -k_d\boldsymbol{\rho}\boldsymbol{\omega} - k_p\boldsymbol{\rho}\boldsymbol{q}_v$ is substituted (the original controller in (34) is the expected control torque, and $\boldsymbol{u}_r$ is the actual torque produced by actuator); under actuator failure, it could be found that

$$\dot{V}_\rho = -\boldsymbol{\omega}^T\boldsymbol{J}\dot{\boldsymbol{\omega}} - 2\boldsymbol{\rho}k_p\dot{q}_0 = -k_d\boldsymbol{\omega}^T\boldsymbol{\rho}\boldsymbol{\omega} \tag{39}$$

Noting that the considered failure in this paper is the drop of control torque but not the positive and negative reverse, $\boldsymbol{\rho}$ is a positive definite matrix, and $\dot{V}_\rho \le 0$ can be ensured. Hence, a system under this condition is still stable. It is worth noting that the proposed method is passive since the coupling matrix $\boldsymbol{\rho}$ is unknown, and the new Lyapunov function is also unknown (but this does not change the controller since the gain factor $\boldsymbol{\rho}$ is not used in controller design). The controller can only ensure system stability, but its performance cannot be ensured.

**Remark 9.** *The fault detection factor $\delta$ can be affected by disturbances such as motor vibration, solar pressure, particle radiation, and so on, and this may cause the mis-judgement of actuator failure. Noting that the norm of these disturbances is at the level of $10^{-4}$–$10^{-5}$ Nm [20,21], which is far less than the torque drop in actuator failure (often $10^{-2}$–$10^{-3}$ Nm), the disturbance would not affect the fault detection algorithm in the situation considered in this paper.*

## 4. Discussion

In this section, numerical simulations for the controller proposed in this paper are conducted to demonstrate its performance.

First, the system parameters are set as follows:

$$\boldsymbol{J} = diag(50, 75, 100)\text{kg} \cdot \text{m}^2, \boldsymbol{d} = 5 \times 10^{-3}\boldsymbol{randn}(3, 1)\text{N} \cdot \text{m}$$
$$\boldsymbol{\omega}(t_0) = \begin{bmatrix} -0.1 & -0.05 & 0.04 \end{bmatrix}^T \text{rad/s}, \boldsymbol{q}_v(t_0) = \begin{bmatrix} \frac{\sqrt{2}}{2} & \frac{\sqrt{3}}{3} & \frac{\sqrt{6}}{6} \end{bmatrix}^T \quad (40)$$
$$t_{sample} = 0.5\text{s}$$

Next, the numerical simulation results would be given.

### 4.1. Comparing Group

In order to demonstrate the performance of the proposed controller, the following standard PD controller was compared.

$$\boldsymbol{u} = -k_d\boldsymbol{\omega} - k_p\boldsymbol{q}_v$$
$$k_d = 10, k_p = 5 \quad (41)$$

The simulation results are given as follows.

Based on the simulation results, it could be found that the system is stable under a standard PD controller. Based on Figures 1 and 2, system convergence time is about 120 s, and steady accuracy is about $6 \times 10^{-4}$ rad/s of angular velocity and $1.5 \times 10^{-3}$ of attitude quaternion. (Generally in a satellite attitude control issue, when the system state approaches its equilibrium point, i.e., $\varphi \approx 0$, we have $\|\boldsymbol{q}_v\| \approx \|\sin \varphi/2\| \approx \varphi/2$. Hence, in this condition, the steady accuracy of the Euler angle is about $3 \times 10^{-3}$ rad, i.e., 0.17 degree, and in later text, this derivation will be omitted.)

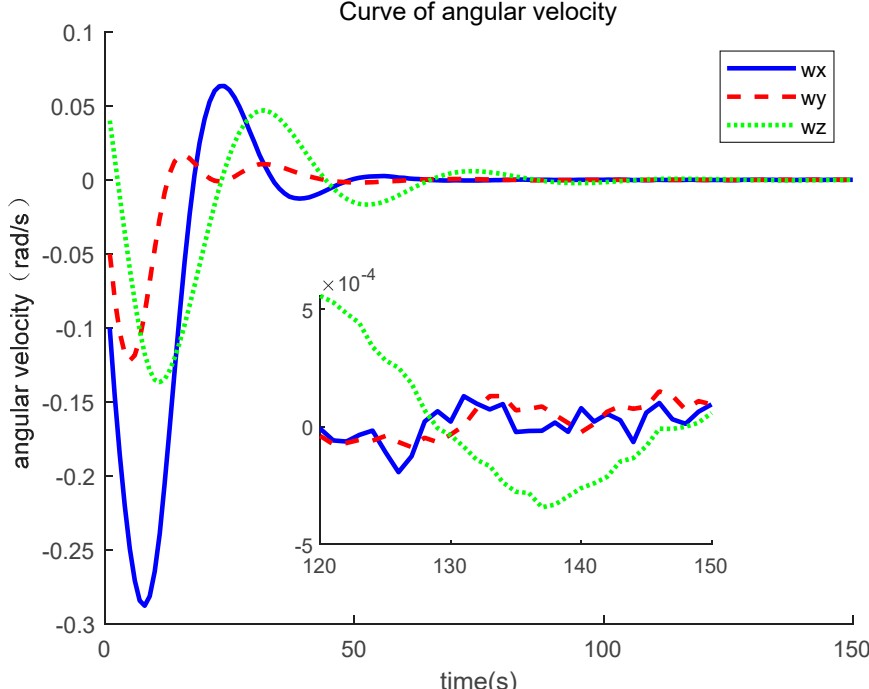

**Figure 1.** Curve of angular velocity under a standard PID controller.

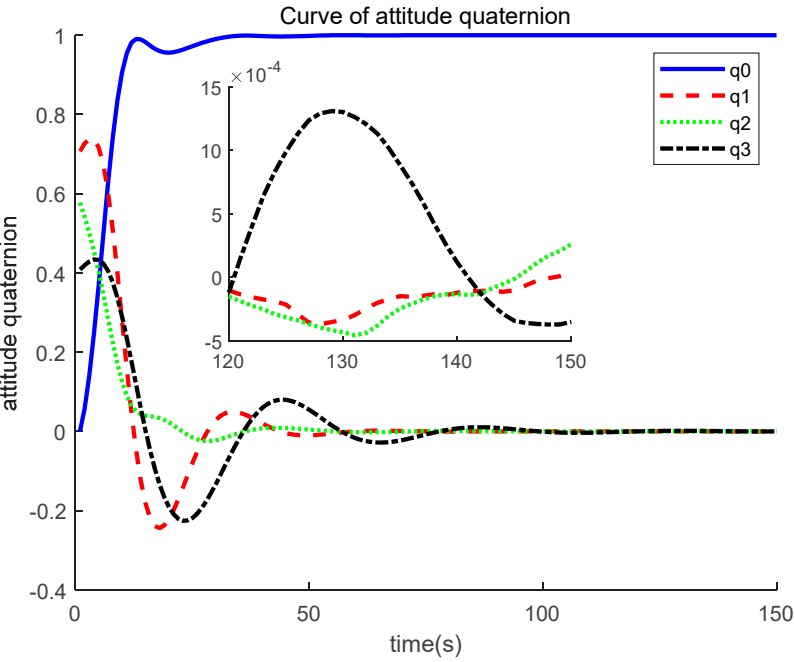

**Figure 2.** Curve of attitude quaternion under a standard PID controller.

The finite time controller proposed from [20] was also compared. The simulation results of [20] are given as follows.

Based on Figures 3 and 4, it could be found that the system convergence rate is about 80 s, which is largely improved compared to a standard PD controller. Steady accuracy is about $6 \times 10^{-6}$ rad/s of angular velocity, $1 \times 10^{-7}$ of attitude quaternion, and $10^{-5}$ degree of the Euler angle. It could also be found that the system has a high frequency issue near the system equilibrium point. The advantages of this method could be concluded as a fast convergence rate and a robustness to model uncertainty, and these were claimed in reference [20].

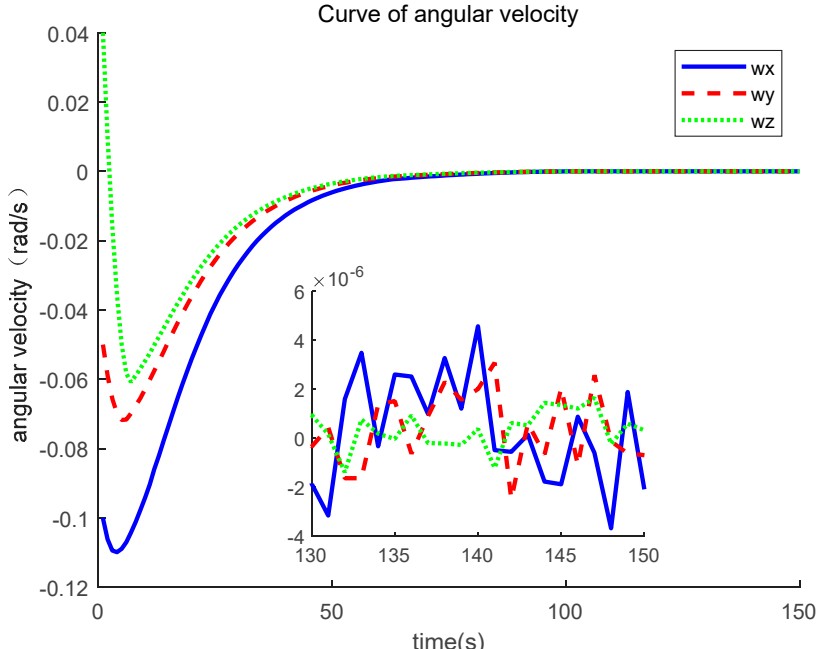

**Figure 3.** Curve of angular velocity under a finite time controller from [20].

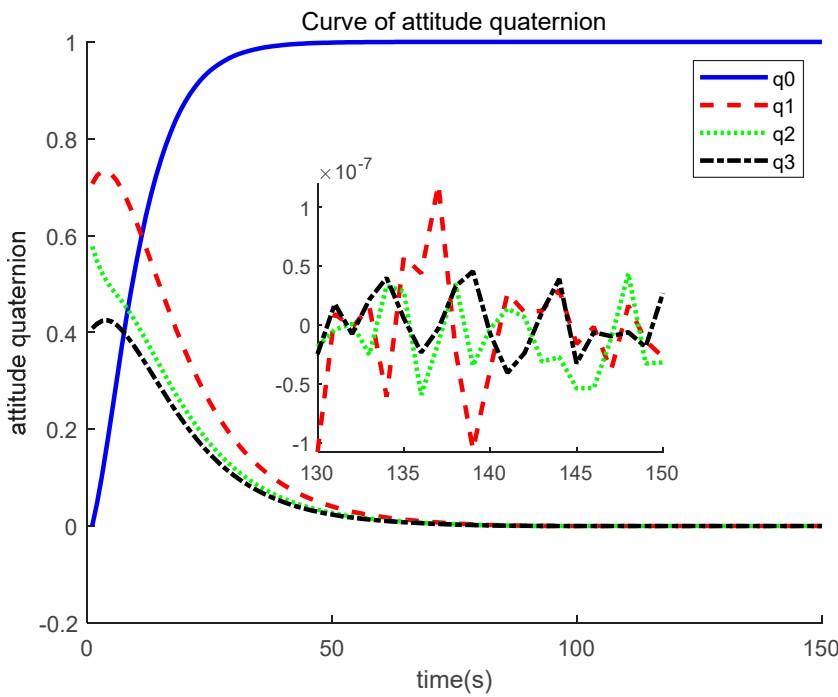

**Figure 4.** Curve of attitude quaternion under a finite time controller from [20].

### 4.2. Variable Structure PID Controller

Next, the simulation results of controller (16) in Section 3.2 are given in this section. The first step is to select the control parameters, and the principle to select control parameters could be concluded as follows: (1) generally a larger differential term could bring a better convergence rate, but the demanded torque is larger and the chattering issue would be aggravated; (2) a larger integral term would bring better robustness to the disturbance, but the system would slow down; (3) a larger proportional term would bring steadier accuracy, but the overshot would be enlarged; (4) the overall control parameters selection could refer to some classic work such as reference [20]. Hence, the control parameters should be selected as follows:

$$k_d = 10, k_p = 5, k_I = 0.2$$
$$\alpha = 0.2, \beta = 0.05, k_1 = 0.1 \tag{42}$$

In addition, in order to check system stability, other Lyapunov function parameters should be selected as follows:

$$l_1 = 0.05, l_4 = 1 \tag{43}$$

Based on constraint (25), it could be calculated that

$$l_2 = l_1 k_d + k_p = 5.5$$
$$l_3 = l_1 k_I + l_4 k_p = 5.001 \tag{44}$$

If (42)–(44) were substituted into constraints (28) and (29), then it could be found that both are satisfied, and the simulation results of controller (16) proposed in this paper are given as follows.

Based on Figures 5 and 6, it could be found that the system is stable under the controller proposed in this paper. The system convergence time is about 50 s, which is largely improved compared to the standard PD controller (almost 40% of the PD controller and 60% of the finite time controller); this verifies that the method proposed in this paper could improve the system convergence rate with a variable structure controller. In addition, the system steady accuracy at 100 s is about $1.5 \times 10^{-5}$ rad/s of angular velocity and $5 \times 10^{-4}$ of attitude quaternion, i.e., 0.01 deg of the Euler angle, which is at the same level compared to the standard PD controller. By comparing the simulations results in this

section and Section 4.1, it could be found that the chattering issue of angular velocity is largely relieved, and this verifies the effectiveness of the integral term. Based on the integral partition algorithm, the integral term does not slow down the system convergence rate; meanwhile, system robustness to the disturbance is maintained.

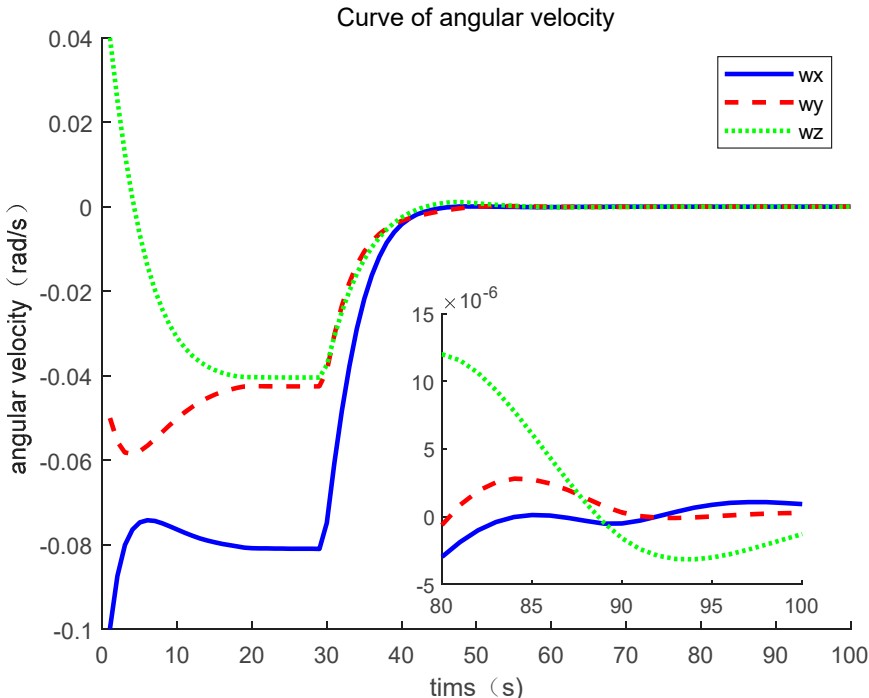

**Figure 5.** Curve of angular velocity under the controller proposed in this paper.

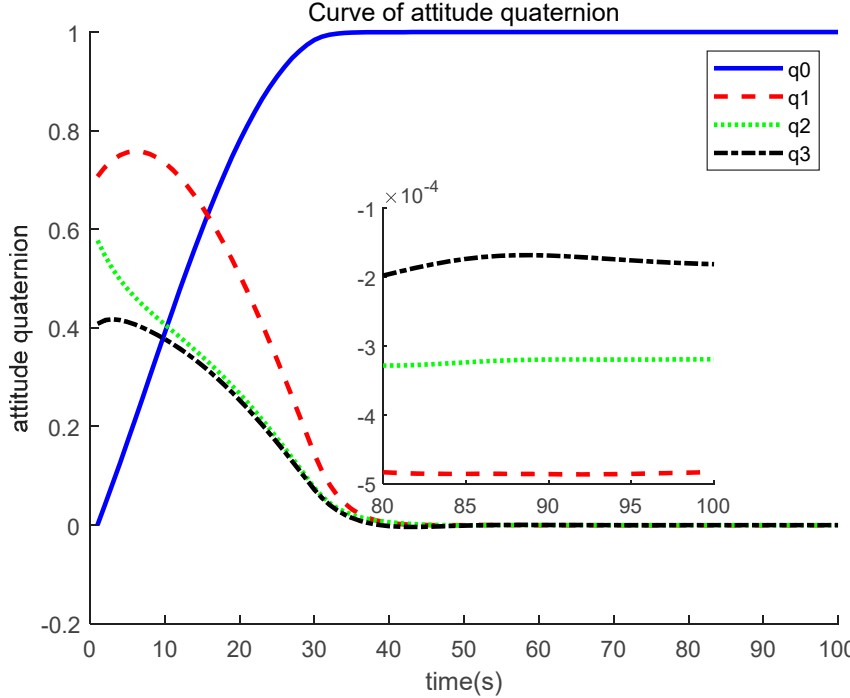

**Figure 6.** Curve of attitude quaternion under the controller proposed in this paper.

### 4.3. Actuator Failure Situation

In this section, the simulation of the fault tolerant controller proposed in Section 3.3 is given. First, assume that, during 20–50 s, the control actuator failure happens, and the system actuator failure is set as follows

$$\boldsymbol{u} = \boldsymbol{\rho}\boldsymbol{u}_d$$
$$\boldsymbol{\rho} = diag(0.5, 0.7, 0.6) \tag{45}$$

It could be found that the actual actuator could only output part of the desired torque in the three axes. Only 50%, 70%, and 60% torque calculated by the control algorithm could be generated by the actuator in the $x, y, z$ axes, respectively. The simulation would be made under this condition for both the controller of the comparing group in Section 4.1 and the controller proposed in this paper.

The fault detection factor $\bar{\delta}$ is set as follows:

$$\bar{\delta} = 5 \times 10^{-3} \tag{46}$$

First, in order to demonstrate the effect of the actuator on the system performance, the PID and finite time controller in Section 4.1 under the same actuator as Equation (45) is given.

Based on Figures 7 and 8, it could be easily found that the system performance is worse than that in Section 4.1, the convergence rate slows down, and the steady accuracy drops. This proves that actuator failure would harm the system performance when it is not properly solved.

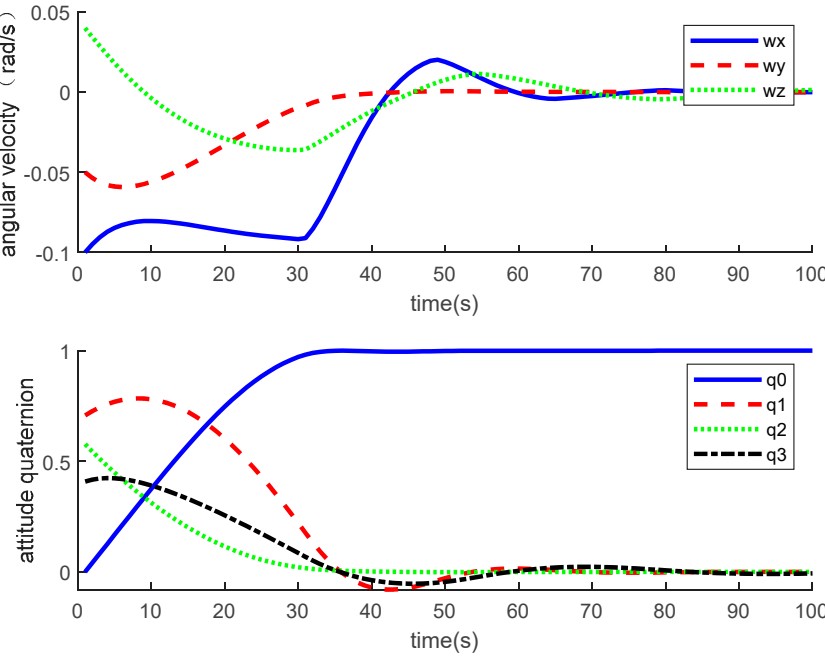

**Figure 7.** Curve of the system state under the standard PID controller with actuator failure.

The simulation results of the fault tolerant controller (34) in Section 3.3 are given as follows.

Based on Figures 9 and 10, it could be found that the system is still stable when the actuator failure happens, and this proves the effectiveness of the method proposed in this paper. The fault detection factor could judge whether the actuator failure happens and when it happens, and the fault tolerant controller could ensure system stability. It could also be found that the system convergence rate is about 80 s, which is slower than the controller in Section 3.2, but still better than the standard PD controller. The system convergence

rate is about $1 \times 10^{-3}$ rad/s of angular velocity and $3 \times 10^{-3}$ of attitude quaternion, i.e., 0.15 deg of the Euler angle, which means that when actuator failure is relieved, the system steady accuracy could still be maintained at a high level. Based on Figure 11 (blue curve is the actual torque and red curve is the expected torque), it could be found that the effect of the actuator is the drop of output torque, and based on the simulation results, it could be found that the proposed method is robust to this type of failure.

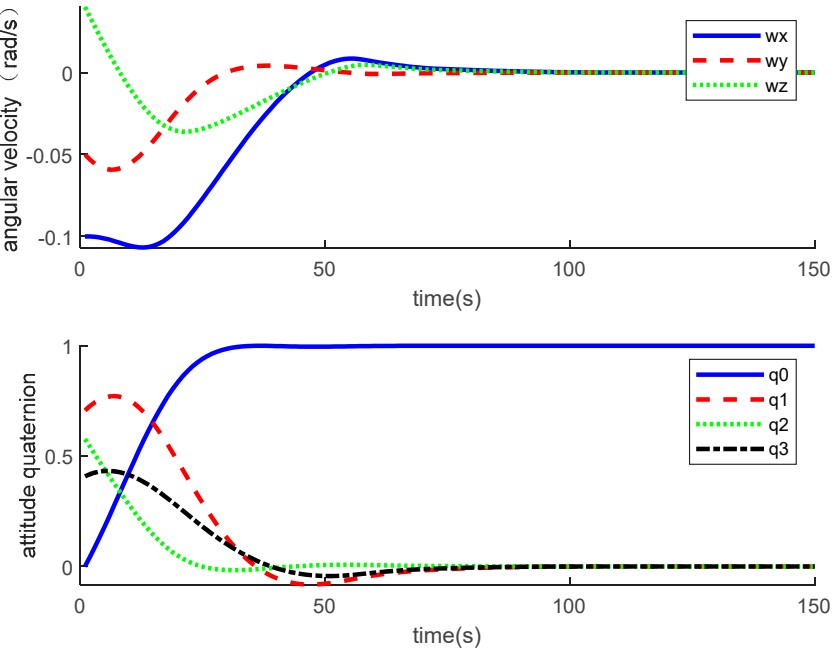

**Figure 8.** Curve of the system state under the finite time controller from reference [20] with actuator failure.

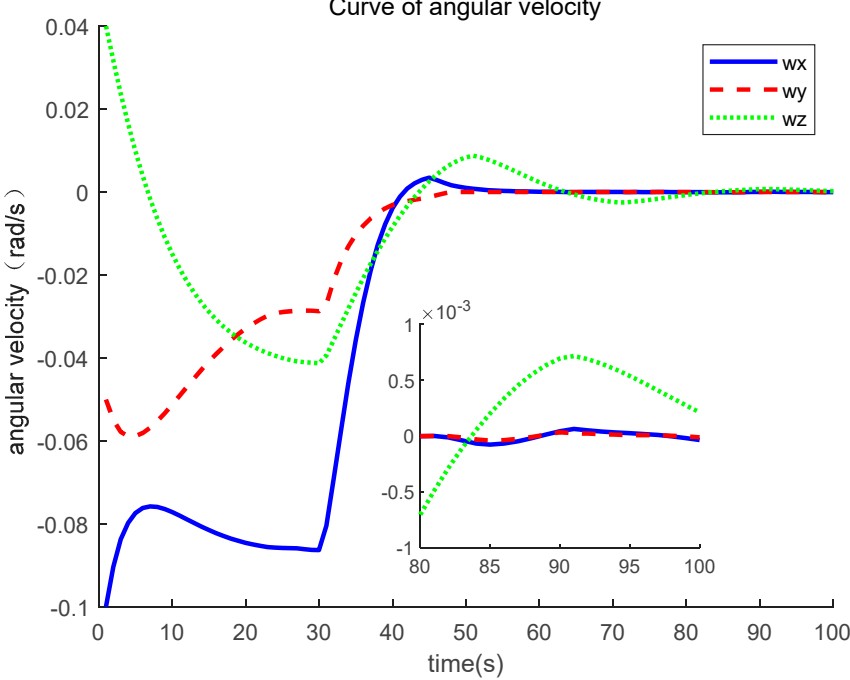

**Figure 9.** Curve of angular velocity under the controller proposed in this paper with actuator failure.

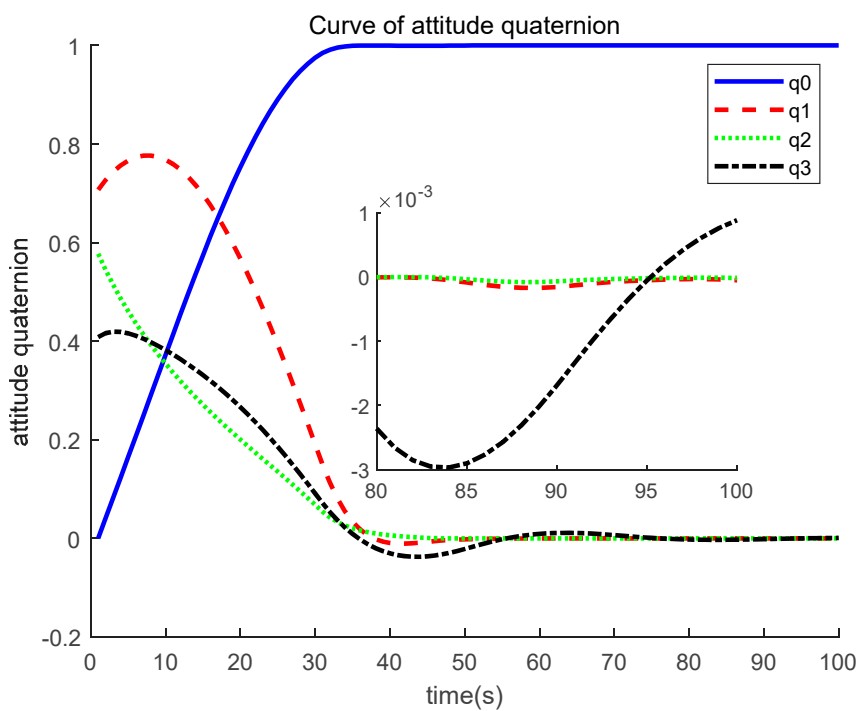

**Figure 10.** Curve of attitude quaternion under the controller proposed in this paper with actuator failure.

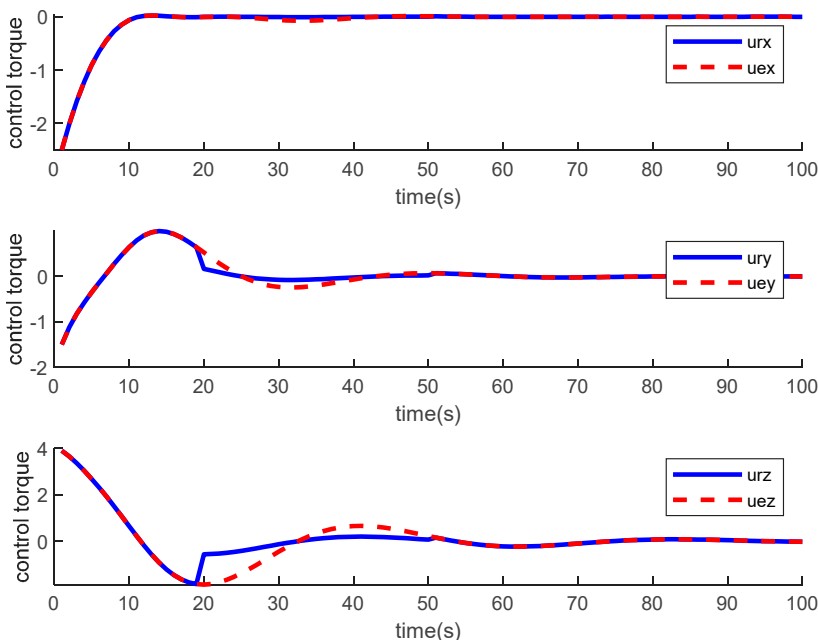

**Figure 11.** Control torque of the proposed controller with actuator failure and without actuator failure.

### 4.4. Summary

The simulation results presented in previous sections could be concluded as follows in Table 1.

Based on the comparison above, it could be found that the proposed method has the best convergence rate and robustness to model uncertainty and disturbance, and the system has a fault tolerant capability when the actuator failure occurs. In addition, the steady accuracy is somehow lower than the finite time controller; however, the chattering issue is solved by the integral term. Hence, it could be concluded that the proposed

method combined the advantages of the standard PD controller and finite time controller; meanwhile, some of the drawbacks of these methods are avoided.

**Table 1.** Comparison of the proposed method and existing methods.

|  | Standard PD Controller | Finite Time Controller | Proposed Method |
|---|---|---|---|
| Convergence time | 120 s | 80 s | 50 s |
| Steady accuracy | $3 \times 10^{-5}$ rad/s (velocity) <br> 0.17 degree (Euler angle) | $6 \times 10^{-6}$ rad/s (velocity) <br> $10^{-5}$ degree (Euler angle) | $1 \times 10^{-5}$ rad/s (velocity $\omega$) <br> 0.01 degree (Euler angle) |
| Robustness | Has high frequency chattering issue near equilibrium point | Has high frequency chattering issue near equilibrium point | Does not rely on accurate model and robustness to disturbance torque |
| Fault tolerant capability | Fault tolerant | None | Fault tolerant |

## 5. Conclusions

In this paper, a variable structure fault tolerant PID controller was proposed. The system performance was improved by the variable structure, integral partition algorithm, and fault tolerant algorithm. The effectiveness and superiority were demonstrated by numerical simulation.

The main contribution of this paper is that a controller combined with PID control, trajectory planning, variable structure, and fault tolerance was proposed; hence, the system has both a good convergence rate and strong robustness. The proposed methods are convenient for engineering practice compared with the methods cited in this paper.

Based on the work of this paper, it could be concluded that: maintaining a high angular velocity is key to improving the system convergence rate, and by implementing a variable structure PD controller, this goal could be achieved; the integral partition algorithm could improve the system performance around its equilibrium point and avoid the issue of slowing down the system convergence rate; the key to solving the actuator is to detect actuator failure by the system state estimation algorithm first and to implement the control algorithm with global stability; the method proposed in this paper has a simple structure; hence, it is convenient for engineering practice, and the complexity to achieve better performance is transformed to stability analysis.

In addition, it is worth noting that the failure of actuators is considered only partly in this paper, and for engineering practice, more actuator failures should be taken into consideration in later work.

**Author Contributions:** Conceptualization, Y.Q.; Data curation, H.J.; Formal analysis, X.W. All authors have read and agreed to the published version of the manuscript.

**Funding:** This research received no external funding.

**Institutional Review Board Statement:** The study was conducted according to the guidelines of the Declaration of Helsinki, and approved by the Institutional Review Board of Shaanxi University of Science and Technology (710021, 2022-04-23).

**Informed Consent Statement:** This study does not involve any informed consent issues since there all the data used in this paper is simulated.

**Data Availability Statement:** If anyone wants the data availability of this paper, please send e-mail to qiyong_sust@163.com.

**Conflicts of Interest:** The authors declare no conflict of interest.

## Nomenclature

| | |
|---|---|
| $J$ | Inertia matrix of satellite ($3 \times 3$ matrix) |
| $\hat{J}$ | Inertia matrix best estimate ($3 \times 3$ matrix) |
| $\tilde{J}$ | Error inertia matrix ($3 \times 3$ matrix) |
| $\boldsymbol{\omega}$ | Angular velocity ($3 \times 1$ vector) |
| $\boldsymbol{u}$ | Control torque ($3 \times 1$ vector) |
| $\boldsymbol{d}$ | Unknown disturbance torque ($3 \times 1$ vector) |
| $\bar{r}$ | Norm upper bound of vector $\boldsymbol{r}$ (scalar) |
| $\boldsymbol{r}^{\times}$ | Product matrix of three-dimensional vector $\boldsymbol{r}$ |
| $\|\boldsymbol{r}\|$ | Euclidean two-norm of vector or matrix $\boldsymbol{r}$ |
| $\boldsymbol{q}$ | Attitude quaternion ($4 \times 1$ vector) |
| $\boldsymbol{q}_v$ | Vector part of attitude quaternion ($3 \times 1$ vector) |
| $q_0$ | Scalar part of attitude quaternion (scalar) |
| $\boldsymbol{e}$ | Euler axis of rigid body rotation ($3 \times 1$ vector) |
| $\varphi$ | Euler angle of rigid body rotation (scalar) |
| $\mathrm{sgn}(x)$ | Sign function of vector or scalar $x$ |
| $\lambda_M(\boldsymbol{A})$ | The maximum eigenvalue of matrix $\boldsymbol{A}$ |
| $\lambda_m(\boldsymbol{A})$ | The minimum eigenvalue of matrix $\boldsymbol{A}$ |

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
