# Peer review of "Variable Structure PID Controller for Satellite Attitude Control Considering Actuator Failure"

_applsci, doi:10.3390/app12105273_

Round 1
Reviewer 1 Report
The paper needs the following revisions for further improvement:
- The abstract and conclusions need improvement. In-text citations also have errors and need improvement. Overall, the literature should be expanded with a state-of-the-art discussion. Please revise the headings/subheadings of the paper (methodology, procedure, results, discussion etc). The paper also requires English proofreading. There are too many grammatical errors. Some sentences are written in the future tense; please revise them.
Author Response
Thank you for your precious opinion to improve this manuscript, and modifications have been made as follows:
(1) About the abstract: We have modified the abstract, and the contributions of this paper is emphasized. The new version of abstract is “In this paper a variable structure PID controller with good convergence rate and robustness for satellite attitude is proposed. In order to improve system convergence rate, the variable structure for proportional and differential term is designed, and angular velocity curve with better convergence rate is achieved by this variable structure. Also, integral partitioning algorithm is designed and system robustness to disturbance torque is improved, meanwhile the negative effect of integral term is avoided during the converging process. The actuator failure condition is also considered and fault tolerant control algorithm is designed. System stability is analyzed by Lyapunov method and its performance is demonstrated by numerical simulation.”.
(2)About the conclusion: We have made some modifications especially the contributions of this paper is added as “The main contribution of this paper is that a controller combined with PID control, trajectory planning, variable structure and fault tolerant is proposed, hence system has both good convergence rate and strong robustness. The proposed methods is convenient for engineering practice comparing with the methods cited in this paper.”.
(4)About the language errors: We have corrected the language errors and the future tense is corrected..
Reviewer 2 Report
This paper presented an interesting variable structure PID controller for controlling satellite flights. After defining the proposed controller, the authors simulated its use to verify its performance, with some comparison done to other similar work. Results show that the controller has a better converge rate and has a fault-tolerant capability. In general, this seems to be a good quality job with interesting results. However, the manuscript should be presented more clearly, with several of the mathematics being more suitable for appendices rather than the main text of the work. In particular, the following points are raised:
- The control accuracy of the fields is relatively low. Would be beneficial to describe quantitatively the state of the art (mm, m, km?).
- Most of the mathematical proof of the PD controller properties (e.g. eq 12, 13, 15) and the addition of the integral term (e.g. eq 22, 23) could be summarized in the main text with part of the derivation perm in an appendix section
- The transformation of equation 28 to allow manual calculations should also be derived in an appendix, with the most simplified form of the constraints (eq 29) presented in the main text
- The zoom-in graph on all figures might be unnecessary considering the level of error that is plotted in those subfigures. Consider removing those.
- The comparison of the proposed PD controller to the work in ref[20] is not clearly presented. The angular velocity and quaternions for both methods should be clearer if presented next to each other with a common axes
- The reasoning behind the definition of the control parameters in section 5.2 is not clear and should be presented
- Figure 7 and 8 should highlight the failure event of the actuator
- The PD controller presented in ref[20] should also be used to tackle the failure challenge and compared vs the results using the proposed controller
- Labels through all of the images are not clearly presented and should be more descriptive. AS an example: Figure 1 should read: Figure 1 Curve of angular velocity simulated based on the PD controller proposed by [20].
Author Response
Thank you for your precious opinion to improve this manuscript, and modifications have been made as follows:
(1) About the accuracy, the control object is satellite attitude in this paper, hence its unit is rad/s of angular velocity and deg of attitude quaternion (Euler angle), and this has been added in the numerical simulation part.
(2)About the proof process, the author has deleted some redundant process to make the paper more concise.
(3)About the zoom-in figure: We use this method to demonstrate the steady accuracy of the controllers, and without the zoom in figure, the steady accuracy is hard to distinguish in the original figures (since the original state is at the level of 100-10-1 but the steady state is at the level of 10-3). Also, the zoom in figure demonstrates the performance near the equilibrium point such as high frequency vibration. Hence we think that it is better to keep the zoom in figures.
(4)About the axis of comparing group: Since the convergence rate of comparing group and the proposed method is not same, that of PID controller, finite time controller and proposed controller is about 120s, 80s and 60s, hence if we set the time axis totally same some of the figures would be congest and some would be wide. Hence in order to make all the text should be used effectiveness, we use different time-varying axis in the text..
(5)About the principle to select control parameters: We have added the principles to select control parameters in section 5.2 (seen as “The first step is to select control parameters and the principle to select control parameters could be concluded as follows: 1.Generally larger differential term could bring better convergence rate, but the demanded torque is larger and the chattering issue would be aggravated; 2.Larger integral term would bring better robustness to disturbance but system would slow down; 3.Larger proportional term would bring better steady accuracy but the overshot also is enlarged; 4.The overall control parameters selection could refer to some classic work such as Ref [20]”).
(6)The control torque under actuator failure is given in Fig.10, and the blue curve demonstrates the actual control torque produced by actuators, and the red curve demonstrates the expected torque calculated by control algorithm.
(7)The controllers in comparing group (standard PID controller and finite time controller) are also tested under same actuator failures in section 5.3. And the simulation results are given and analyzed in section 5.3.
(8)All the labels of the figures have been corrected throughout the paper.
Reviewer 3 Report
- General comments:
The authors proposed a variable structure fault tolerant PID controller. They improved the system performance by using this variable structure and by using an integral partition algorithm as well as a fault tolerant algorithm. Simulation results showed the effectiveness and superiority compared to similar systems.
- Analysis:
Quality of the writing: structured coherently, some English mistakes should be corrected.
Abstract: Easy to understand
Introduction: well written, context is clear to some extent.
Problematic: clearly identified
Method: well-described
Application field: well-identified
Results: well-illustrated.
Conclusion: concise
Related works: incomplete and not well structured
Originality: very incremental
3. Specific comments:
The related works should be completed. For example, the recent works of You Li are missing:
https://www.hindawi.com/journals/mpe/2021/5539717/
https://www.mdpi.com/2073-8994/14/1/45/pdf
Author Response
Thank you for your precious opinion to improve this manuscript, and modifications have been made as follows:
(1)The recent related works have been cited in the text (Ref 20-22), and its contribution and drawbacks have been compared with the proposed method (Seen in section 1 as “Also, Li[20-22] designed satellite attitude controllers for fast attitude maneuver but the actuator failures are not taken into consideration.”).
Round 2
Reviewer 2 Report
I would like to thank the authors for their modifications. I believe this paper is now in a form which warrants publication and I have no major comments further.
Author Response
Thanks for the precious opinions and suggestions to make this manuscript better.
Reviewer 3 Report
Quality of the writing: structured coherently.
Abstract: Easy to understand
Introduction: well written, context is clear to some extent.
Problematic: clearly identified
Method: well described
Application field: well identified
Results: well illustrated.
Conclusion: short and contains concrete conclusions
Reviewers:
The reviewers did an excellent job.
They raised important points.
They asked pertinent questions.
Reply to the reviews:
The authors replied adequately to the reviewers.
Author Response
Thanks a lot for the precious opinions and suggestions to make ths manuscript better.